# Engineering *Pseudomonas putida* KT2440 for chain length tailored free fatty acid and oleochemical production

Luis E. Valencia[1,2,3], Matthew R. Incha[1,2,4], Matthias Schmidt[1,2,5], Allison N. Pearson[1,2,4], Mitchell G. Thompson [1,6], Jacob B. Roberts[1,2,3], Marina Mehling[1,2], Kevin Yin [1,2,4], Ning Sun[2,7], Asun Oka[2,7], Patrick M. Shih[1,2,4,6], Lars M. Blank [5], John Gladden[1,8] & Jay D. Keasling [1,2,3,9,10,11]✉

Despite advances in understanding the metabolism of *Pseudomonas putida* KT2440, a promising bacterial host for producing valuable chemicals from plant-derived feedstocks, a strain capable of producing free fatty acid-derived chemicals has not been developed. Guided by functional genomics, we engineered *P. putida* to produce medium- and long-chain free fatty acids (FFAs) to titers of up to 670 mg/L. Additionally, by taking advantage of the varying substrate preferences of paralogous native fatty acyl-CoA ligases, we employed a strategy to control FFA chain length that resulted in a *P. putida* strain specialized in producing medium-chain FFAs. Finally, we demonstrate the production of oleochemicals in these strains by synthesizing medium-chain fatty acid methyl esters, compounds useful as biodiesel blending agents, in various media including sorghum hydrolysate at titers greater than 300 mg/L. This work paves the road to produce high-value oleochemicals and biofuels from cheap feedstocks, such as plant biomass, using this host.

[1] Joint BioEnergy Institute, Emeryville, CA 94608, USA. [2] Biological Systems and Engineering Division, Lawrence Berkeley National Laboratory, Berkeley, CA 94720, USA. [3] Department of Bioengineering, University of California, Berkeley, CA 94720, USA. [4] Department of Plant and Microbial Biology, University of California, Berkeley, CA 94720, USA. [5] Institute of Applied Microbiology (iAMB), Aachen Biology and Biotechnology (ABBt), RWTH Aachen University, Aachen, Germany. [6] Environmental Genomics and Systems Biology Division, Lawrence Berkeley National Laboratory, Berkeley, CA, USA. [7] Advanced Biofuels and Bioproducts Process Demonstration Unit, Emeryville, CA 94608, USA. [8] Biomanufacturing and Biomaterials Department, Sandia National Laboratories, Livermore, CA 94550, USA. [9] Department of Chemical & Biomolecular Engineering, University of California, Berkeley, CA 94720, USA. [10] Center for Biosustainability, Technical University of Denmark, Lyngby, Denmark. [11] Center for Synthetic Biochemistry, Institute of Synthetic Biology, Shenzhen Institutes of Advanced Technologies, Shenzhen, China. ✉email: keasling@berkeley.edu

The world has seen a monumental shift to oleochemicals, plant and animal-derived oils and fats, as a substitute for petrochemicals in recent decades[1,2]. These oleochemicals are a critical renewable feedstock for the industrial production of detergents, lubricants, and biodiesel, among other products[3]. Palm oil has emerged as the dominant and fastest-growing source of oleochemicals due to its superior productivity of ~4.4 metric tonnes per hectare per year[1]. However, there is demand for alternative sources of oleochemicals that rely less on sensitive tropical land use and can provide fatty acids with different physicochemical properties such as shorter chain lengths (<C16), branched chains, and unique stereochemistry[3,4]. Currently, most biodiesel is derived from plants and results in predominantly long-chain (C16-C18) fatty acid methyl esters (FAMEs). These long-chain FAMEs have favorable cetane numbers, however, they show poor performance at low temperatures. Consequently, to be used as drop-in fuels, they require blending agents such as short- or branched-chain FAMEs[5,6]. Metabolic engineering of microbes can provide a route to unique oleochemicals that are difficult to obtain from plants & animals, and may have favorable fuel characteristics[7–10].

*Pseudomonas putida* KT2440 is a promising host for industrial oleochemical production due to its remarkable ability to utilize recalcitrant carbon sources found in lignocellulosic biomass[11,12], high tolerance to oxidative stress[12], strong redox potential[13], and favorable growth characteristics[14]. Many genetic tools have become available for engineering *P. putida*[15–17], and substantial progress has been made in improving its host properties, such as enhanced fitness in bioreactors[18], utilization of plant-derived hydrolysates[19], and tolerance for ionic liquids found in these hydrolysates[20]. Although *P. putida* has shown itself to be a versatile and robust host for bioproduction[18,21–23], to date there has not been a reported strain capable of producing free fatty acids (FFAs), which are important intermediates in oleochemical metabolic pathways as well as valuable precursors for industrial processes[24].

To produce FFAs, the host organism must be unable to catabolize FFAs via β-oxidation, and an acyl-ACP thioesterase must be employed to hydrolyze and offload FFAs from fatty acid biosynthesis[24,25]. Avoiding β-oxidation of FFAs can be achieved by knocking out fatty acyl-coenzyme A (CoA) ligases responsible for initiating β-oxidation. In contrast to *Escherichia coli*, which only has two CoA ligases involved in β-oxidation[26], *P. putida* was recently found to have a much larger repertoire (>4) of CoA ligases involved in FFA degradation. Growth experiments with individual deletion strains of these CoA ligases suggested that they are involved in initiating β-oxidation of FFAs and have different substrate preferences with regard to the carbon chain length of the FFA[27].

In this work, we report the development of a *P. putida* strain unable to catabolize medium and long-chain FFAs by knocking out three CoA ligases. This triple knockout strain produced a different profile of FFAs depending on the heterologous thioesterase expressed, achieving a titer of 670 mg/L total FFAs in shake flask culture. A double knockout strain that favors the production of medium-chain FFAs was developed by leveraging the in vivo substrate preferences of the CoA ligases (Fig. 1). This strategy can potentially be employed in other FFA-producing microbes. Finally, a heterologous fatty acid methyltransferase was expressed to generate medium-chain FAMEs, achieving a titer of 302 mg/L total FAMEs in shake flask cultures.

## Results

### Engineering P. putida to avoid catabolism of medium- and long-chain FFAs.
Previously, by using an RB-TnSeq mutant library of *P. putida*, our lab identified several mutants with moderate to severe fitness defects when grown on minimal media containing FFAs as a sole carbon source[27]. Transposon mutants in CoA ligase genes *PP_0763*, *PP_4549*, and *PP_4550* had particularly strong fitness defects when grown on medium- (C6-C12) and long-chain (C14≤) fatty acids. Notably, the strength of the fitness defect varied depending on the mutant and the FFA used. *PP_0763* had a severe fitness defect when grown on C5-C6 FFAs and a moderate defect when grown on C7-C10 FFAs. *PP_4549* had a severe fitness defect when grown on C6-C18 FFAs, and *PP_4550* had a moderate fitness defect when grown on C6-C12

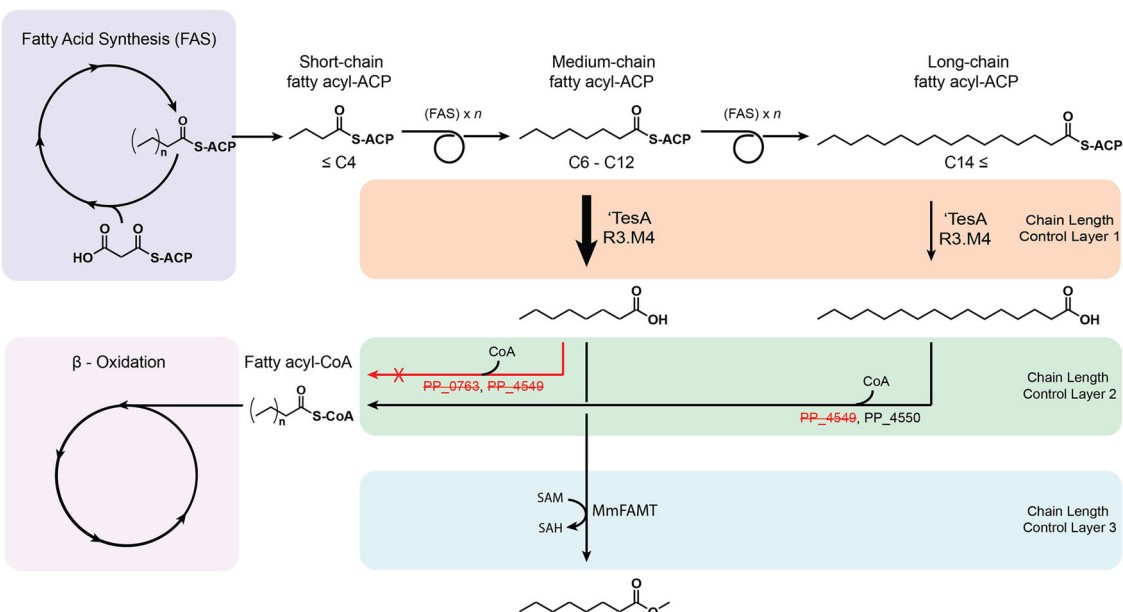

**Fig. 1 Multi-layer chain length control strategy.** The chain length of the final product is influenced by three layers of control. The use of an engineered acyl-ACP thioesterase favors the production of medium-chain FFAs in layer 1. Selectively knocking out CoA ligases with medium-chain FFA substrate preference prevents the catabolism of medium-chain FFAs while still allowing for catabolism of long-chain FFAs in layer 2. The use of a medium-chain fatty acid methyltransferase further influences the final chain-length profile in layer 3.

FFAs (Supplementary Fig. 1). In this work, we introduced single, double, and triple knockout combinations of these CoA ligase genes and assessed the ability of these mutant strains to catabolize medium- and long-chain FFAs.

We conducted FFA degradation assays with the constructed mutant strains, in which we back diluted overnight cultures into EZ Rich medium containing 10 mM glucose and 250 μM of each straight-chained FFA with an even-numbered chain length between C6-C16. After a 48-h growth period, FFAs were derivatized & extracted from cultures, quantified via GC/MS, and compared to a media-only control (Fig. 2). Wild-type *P. putida* and single knockouts of ΔPP_0763 and ΔPP_4549 were found to degrade more than 80% of each added FFA. The double knockout strains, though still proficient at degrading FFAs, were found to preferentially degrade certain chain lengths depending on which CoA ligases were knocked out. ΔPP_0763 ΔPP_4549 degraded more than 80% of each FFA except C6, while ΔPP_4549-50 degraded more than 80% of each FFA except C14 and C16. This reflects the hypothesis that these CoA ligases have different, yet overlapping, substrate preferences and particular combinations of CoA ligase knockouts may preferentially degrade medium- or long-chain FFAs. The triple knockout strain, ΔPP_0763 ΔPP_4549-50, hereinafter referred to as 3KO, was found to not degrade any FFAs except C6 and in fact produced small amounts of C8, C12, C14, and C16, likely due to the activity of an endogenous thioesterase. 3KO was able to degrade approximately 30% of the added C6 FFA which may be explained by the activity of PP_3553, which is predicted to be a short-chain CoA ligase and was reported have a slight fitness defect when grown on hexanoic acid as a sole carbon source[27]. Finally, 3KO was unable to grow on medium-chain FFAs as a sole carbon source (Supplementary Fig. 2), which further demonstrates that this strain is unable to catabolize medium-chain FFAs via β-oxidation.

**Engineering *P. putida* to produce medium- and long-chain FFAs.** Following the construction of 3KO, a strain unable to catabolize medium- and long-chain FFAs, we sought to construct strains that actively produce FFAs by overexpressing an acyl-ACP thioesterase. We compiled five engineered variants of 'TesA, a leaderless version of acyl-ACP thioesterase I from *E. coli*[28] (Table 1) and overexpressed each in both wild-type and 3KO background *P. putida* strains via an arabinose-inducible pBADT vector[29] (Fig. 3). After a 48-h growth period in either EZ Rich supplemented with 100 mM glycerol or Terrific Broth (TB), FFAs were derivatized and extracted from cultures and quantified via GC/MS. All wild-type background strains produced virtually no FFAs, while all 3KO background strains produced FFAs. All 'TesA variants produced similar FFA profiles to what was reported in the literature (Table 1) in both EZ Rich and TB media. 3KO containing a pBADT-RFP control plasmid was also found to produce FFAs, primarily C16 and monounsaturated C16 (C16:1), which is likely due to an endogenous thioesterase. Notably, the R3.M4 'TesA variant produced the highest titer in both media and produced primarily medium-chain (C6-C12) FFAs. R3.M4 was selected as the 'TesA variant of choice for future experiments. Interestingly, 3KO strains containing a 'TesA variant produced isovalerate when grown on TB, perhaps as a degradation product of leucine found in the medium[30] (Supplementary Fig. 3).

Given the apparent differences in CoA ligase substrate preference, we hypothesized that expressing R3.M4 in single or double knockout strains would enrich specifically for either medium- or long-chain FFAs because the remaining CoA ligase(s) would still initiate β-oxidation of their preferred FFAs. To achieve this, plasmid 115 was introduced into wild-type, single, double, and triple knockout strains, and FFAs were derivatized and quantified via GC/MS after a 48-h growth period in either EZ Rich supplemented with 100 mM glycerol or TB

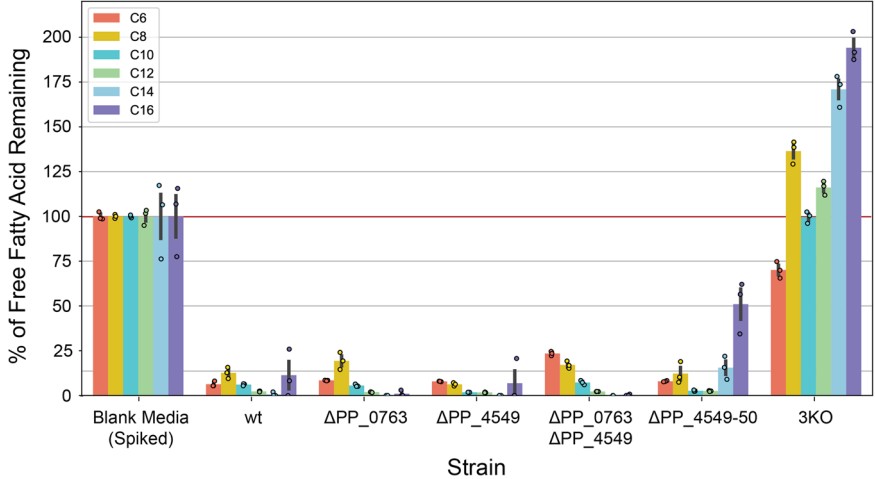

**Fig. 2 FFA degradation assays.** FFAs concentrations quantified 48 h after strain growth in media spiked with 250 μM of C6-C16 even-chain FFAs and normalized by FFA concentration in the original spiked media. Values shown represent the mean of biologically independent samples (*n* = 3), and error bars show standard error of the mean.

**Table 1 Details of the 'TesA variants and their pBADT plasmid names used in this study.**

| 'TesA variant | Plasmid name (this study) | Mutation | Major FFA produced (*E. coli*) | Ref. |
|---|---|---|---|---|
| 'TesA | 113 | N/A | C16 | 28 |
| L109P | 114 | L109P | C14, C16 | 52 |
| R3.M4 | 115 | M141L, Y145K, L146K | C8 | 40 |
| CM-5 | 116 | E142D, Y145G | C12/C14 | 53 |
| RD-2 | 117 | M141L, E142D, Y145G, L146K | C8 | 53 |

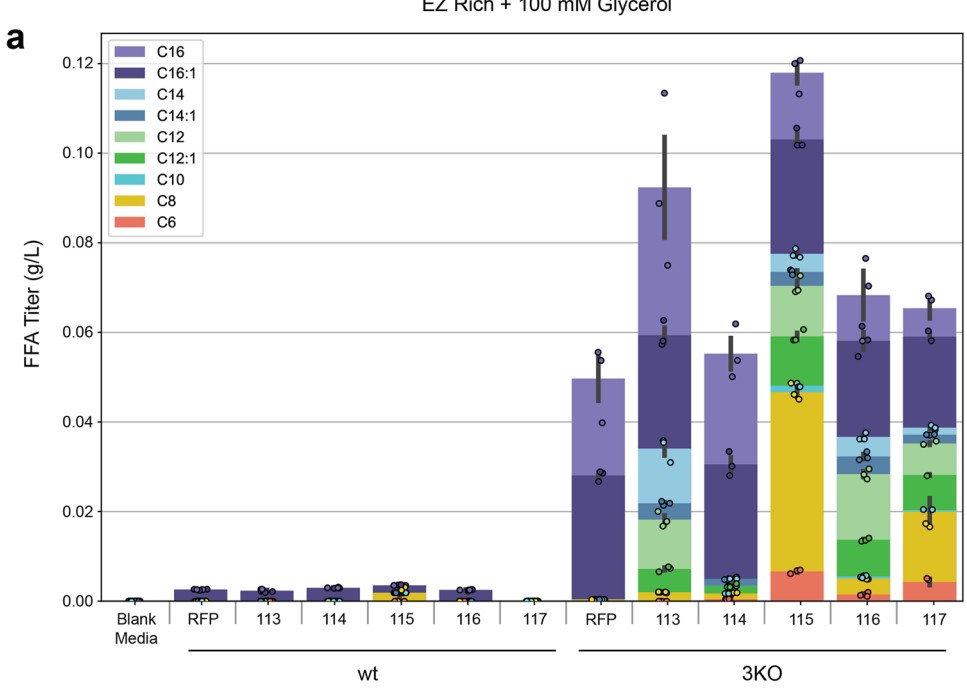

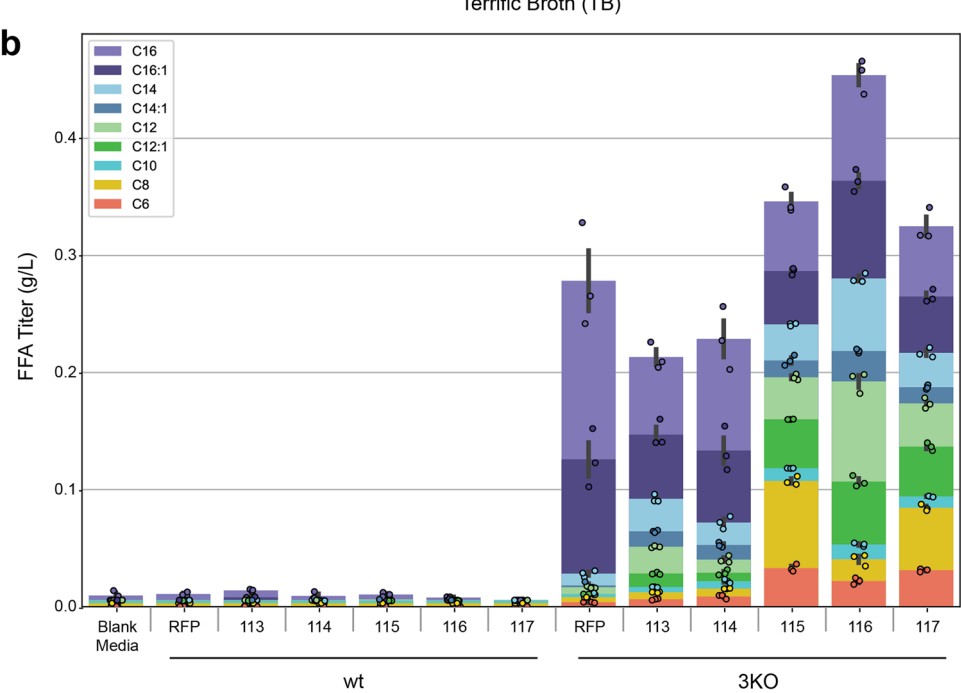

**Fig. 3 FFA production profile across various 'TesA variants in wt and 3KO background strains. a** FFA production in EZ Rich medium supplemented with 100 mM glycerol. **b** FFA production in TB. Values shown represent the mean of biologically independent samples ($n = 3$), and error bars show standard error of the mean.

(Fig. 4). In EZ rich, aside from the 3KO strains which performed similarly to what was reported in Fig. 3, the only strain able to produce a substantial amount of FFAs was Δ*PP_0763* Δ*PP_4549* + 115, which almost exclusively produced medium-chain FFAs. This result supports the hypothesis that *PP_4550* prefers long-chain FFA substrates; consequently, Δ*PP_0763* Δ*PP_4549* is still able to catabolize long-chain FFAs while producing medium-chain FFAs. The same pattern was observed in TB, Δ*PP_0763* Δ*PP_4549* + 115 almost exclusively produced

medium-chain FFAs. TB is considerably more nutrient rich than EZ Rich, and thus in TB we observed higher titers from all strains, including the double knockout strains Δ*PP_0763* Δ*PP_4549* and Δ*PP_4549-4550* containing the negative control RFP plasmid. FFA production in RFP-containing strains likely results from endogenous thioesterase activity; Δ*PP_0763* Δ*PP_4549* + RFP produced mostly medium-chain FFAs while Δ*PP_4549-4550* + RFP produced long-chain FFAs. The single knockout and double knockout strains containing plasmid 115 also

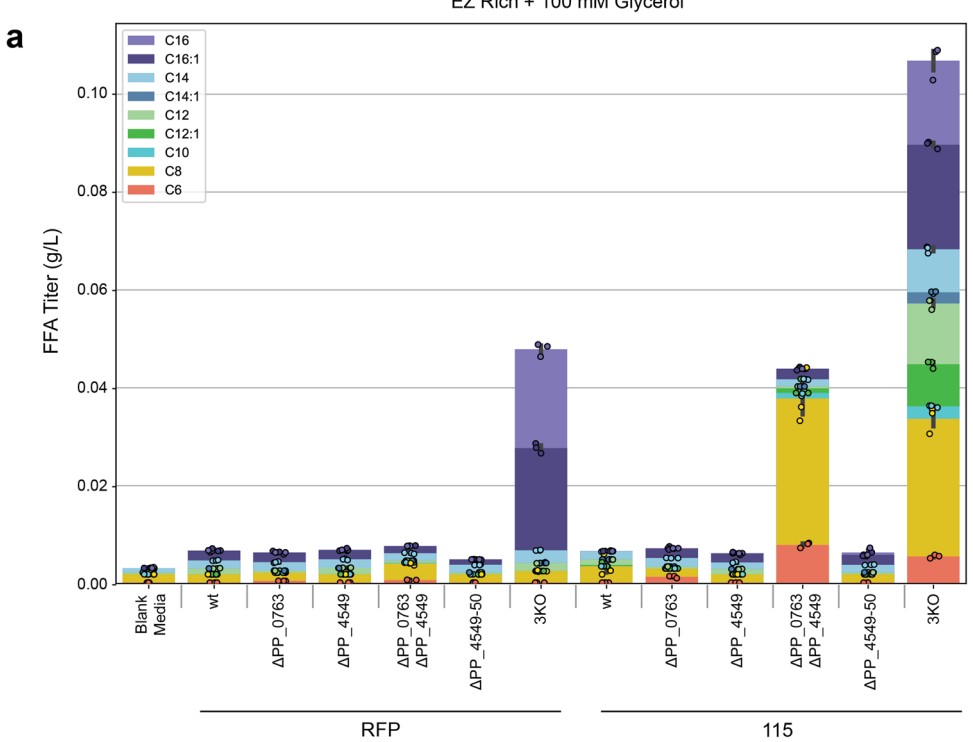

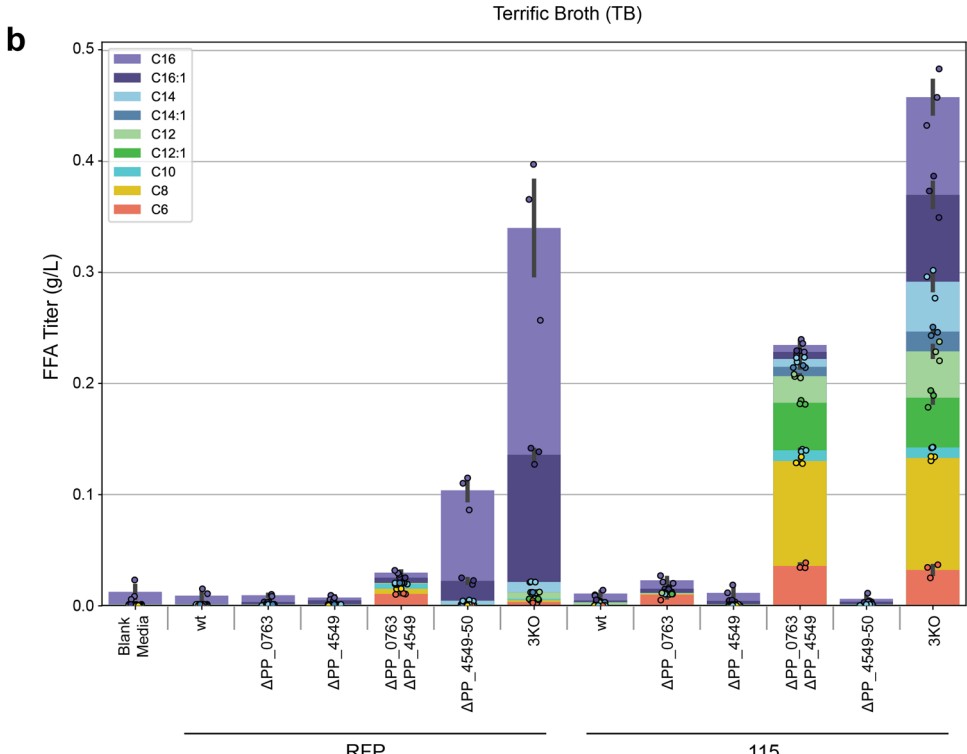

**Fig. 4 FFA production profile across various *P. putida* background strains after overexpression of RFP or 'TesA R3.M4. a** FFA production in EZ Rich medium supplemented with 100 mM glycerol. **b** FFA production in TB. Values shown represent the mean of biologically independent samples ($n = 3$), and error bars show standard error of the mean.

produced up to 109.2 mg/L of isovalerate when grown on TB (Supplementary Fig. 3).

**Engineering *P. putida* to produce FFA-derived FAMEs.** To showcase the ability of FFA-producing *P. putida* strains to synthesize FFA-derived oleochemicals, we aimed to produce FAMEs by using a fatty acid methyltransferase from *Mycobacterium marinum*, MmFAMT. This methyltransferase works by transferring a methyl group from S-adenosylmethionine (SAM) to a fatty acid carboxyl group and was found to be active on C8, C10, and C12 saturated FFAs and their respective 3-hydroxyacids[31,32]. MmFAMT was previously used to produce FAMEs in *E. coli*, though the highest titers were relatively low, at 16 mg/L, likely because most of the FFAs produced in those *E. coli* strains were long-chain FFAs[32]. We expected MmFAMT to perform similarly in the *P. putida* knockout strains that favor the production of medium-chain FFAs, the preferred substrate of MmFAMT. To test this, we constructed plasmid 125 from plasmid 115, inserting the engineered *P. putida* RBS JER05[33] and MmFAMT immediately downstream of R3.M4. This plasmid was introduced into the wild-type, single, double, and triple knockout strains. Strains containing plasmid 125 were grown using the same protocol used previously in this study to quantify FFA production. To reduce evaporation of the FAMEs, a hydrophobic isopropyl myristate (IPM) overlay was added to each culture. IPM was used as the overlay because it demonstrates low toxicity to *P. putida* (Supplementary Fig. 4), is effective at extracting FAMEs, and does not interfere with the chromatography of the FAMEs due to its non-overlapping retention time. At the end of the experiment, this overlay was collected and diluted with ethyl acetate containing an internal standard, and FAMEs were quantified via GC/MS. To account for partial evaporation, blank media samples with an IPM overlay were spiked with 500 μM of each medium-chain FAME and incubated alongside FAME-producing cultures. The final measured concentration of FAMEs in these spiked blank media samples was used to calculate an evaporation factor that was applied to experimental FAME measurements. The calculated evaporation factor varied depending on the media used, the size of the culture tube or flask, and the chain-length of the FAME (Supplementary Table 1).

In both EZ Rich and TB media, the production of FAMEs was observed in all strains containing 125, though the titers varied drastically based on the background strain (Fig. 5). The only FAMEs observed were C6, C8, C10, C12:1, and C12, which confirms the medium-chain substrate preference of MmFAMT. All strains grown in EZ Rich produced low titers of FAMEs, with the highest titer of 28.1 mg/L being achieved by wild-type + 125. In contrast, when grown in TB, wild-type + 125 was the lowest producing strain whereas the highest producing strains were ΔPP_0763 ΔPP_4549 and 3KO, with titers of 192.5 mg/L and 163.8 mg/L, respectively. This drastic difference in titers achieved in EZ Rich and TB media is possibly explained by the fact that the methylation step is dependent on the availability of the SAM cofactor, which is produced from methionine and ATP precursors that are more readily available/produced from a very rich medium such as TB. A decreased ability to produce SAM in strains grown in EZ Rich may also explain why the wild-type background strain produced the highest FAMEs titer in this medium; the wild-type strain grows comparatively faster than 3KO (Supplementary Fig. 4) and is also able to recycle FFAs as a source of carbon and energy which may help it slightly improve its SAM availability and result in more production of FAMEs that are then sequestered into the IPM overlay.

FFA production in strains containing 125 was also measured (Fig. 5). In general, we observed the same FFA production patterns as in previous experiments, but the effect of CoA ligase substrate preference was amplified for FFAs that could be methylated by MmFAMT. For example, the single knockout ΔPP_0763 + 125 produced almost exclusively C6, C8, and C12:1 FFA in both media. In contrast ΔPP_0763 + 115 did not produce appreciable amounts of any FFA. Considering that MmFAMT competes with native CoA ligases for medium-chain FFAs, it is likely that methylation renders a FAME unavailable for CoA ligases. Therefore, in a strain missing a CoA ligase specialized in C6-C12 FFAs, such as ΔPP_0763, the MmFAMT is more likely to outcompete the remaining CoA ligases and produce C6-C12 FAMEs. In cultures containing an overlay, these FAMEs are sequestered into the overlay. In cultures without an overlay, the majority of the FAMEs evaporate, otherwise they remain in solution or hydrolyze (either spontaneously or by the action of a methyl ester esterase such as *bioH*)[34]. FAMEs that remain in solution or are hydrolyzed will both result in an elevated FFA measurement. Similarly, ΔPP_4549 + 125 produced primarily C12:1 and C12 FFAs, while ΔPP_4549 + 115 did not produce appreciable amounts of FFAs. ΔPP_0763 ΔPP_4549 + 125 produced exclusively medium-chain FFAs with a particular abundance of C8. This finding is expected since the sample's background strain cannot catabolize medium-chain FFAs, possesses a thioesterase specialized in making medium-chain FFAs, and contains a methyltransferase with a medium-chain FFA substrate preference.

**Shake flask production runs and media comparisons.** To further assess the potential for *P. putida* to produce FFAs and FFA-derived oleochemicals, we conducted production runs of select strains using EZ Rich + 100 mM glycerol, TB, and 4x diluted sorghum hydrolysate. Sorghum hydrolysate is a rich medium derived from plant lignocellulosic biomass and is attractive due to its industrial relevance as a carbon-neutral and plant-derived feedstock for the production of biofuels and bioproducts[35]. Following dilution, the hydrolysate contained 11.8 g/L glucose, 6.4 g/L xylose, 4.2 g/L acetic acid, 4.3 g/L lactic acid, and unquantified amounts of choline and lignin monomers which *P. putida* can use as sources of carbon and energy[36]. Due to the high sugar and ionic liquid content of sorghum hydrolysate, strains were gradually adapted to grow in diluted media (see methods). Since fatty acid biosynthesis is an energy and redox-intensive process[13], these production runs were conducted at a volume of 12.5 mL in 250-mL unbaffled shake flasks to provide optimal aeration. Growth in unbaffled flasks increased titers relative to 5 mL culture tubes across all samples; results are summarized in Fig. 6 and Supplementary Tables 2 and 3. Additionally, yield calculations were conducted for strains grown in EZ Rich + 100 mM glycerol and are summarized in Supplementary Tables 4–7.

Notably, C8 FFA and C8 FAME were the major products in all strains containing 115 and 125, respectively. This reflects the FFA profile produced by the R3.M4 thioesterase. Additionally, ΔPP_0763 ΔPP_4549 strains were enriched in medium-chain FFAs compared to 3KO strains, which had a more even distribution of medium- and long-chain FFAs. Interestingly, although FAME production improved across all experiments in shake flasks, it did not have a clear correlation with FFA titers. This suggests that although it is important to have a supply of medium-chain FFAs for MmFAMT to utilize, cryptic regulatory or metabolic flux influences might also affect final titers. The highest total FFA titer achieved was 670.9 mg/L, produced by 3KO + 115 in TB. The highest C8 FFA titer, 253.6 mg/L, was observed when ΔPP_0763 ΔPP_4549 + 115 was grown in TB. The greatest FAME titer was also achieved in TB; ΔPP_0763 ΔPP_4549 + 125 in TB produced 302.4 mg/L total FAMEs.

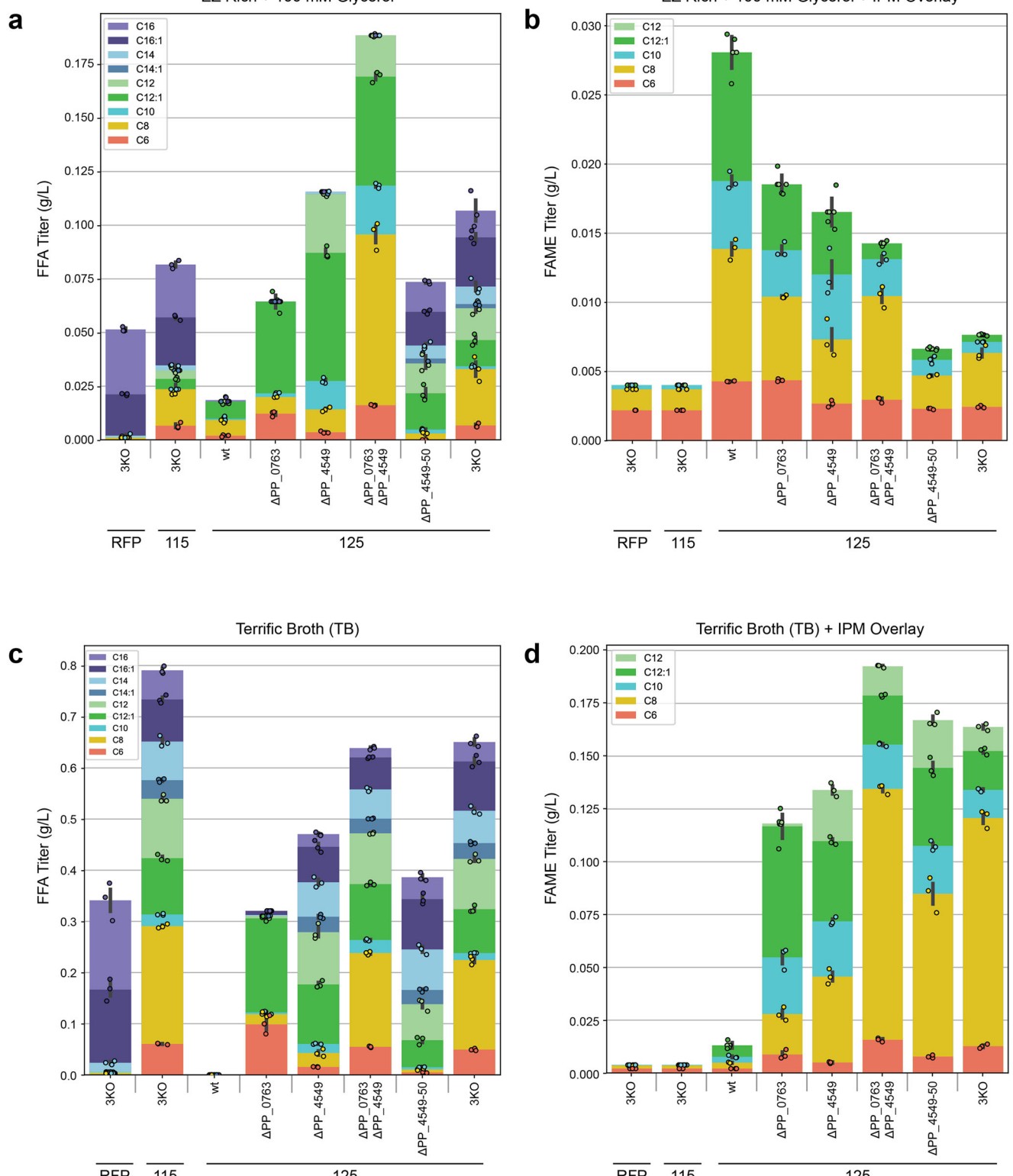

**Fig. 5 FFA and FAME production profile across various *P. putida* background strains after overexpression of RFP, 'TesA R3.M4 only, or 'TesA R3.M4 & MmFAMT. a** FFA production in EZ Rich medium supplemented with 100 mM glycerol. **b** FAME production in EZ Rich media supplemented with 100 mM glycerol. **c** FFA production in TB. **d** FAME production in TB. Values shown represent the mean of biologically independent samples (*n* = 3), and error bars show standard error of the mean.

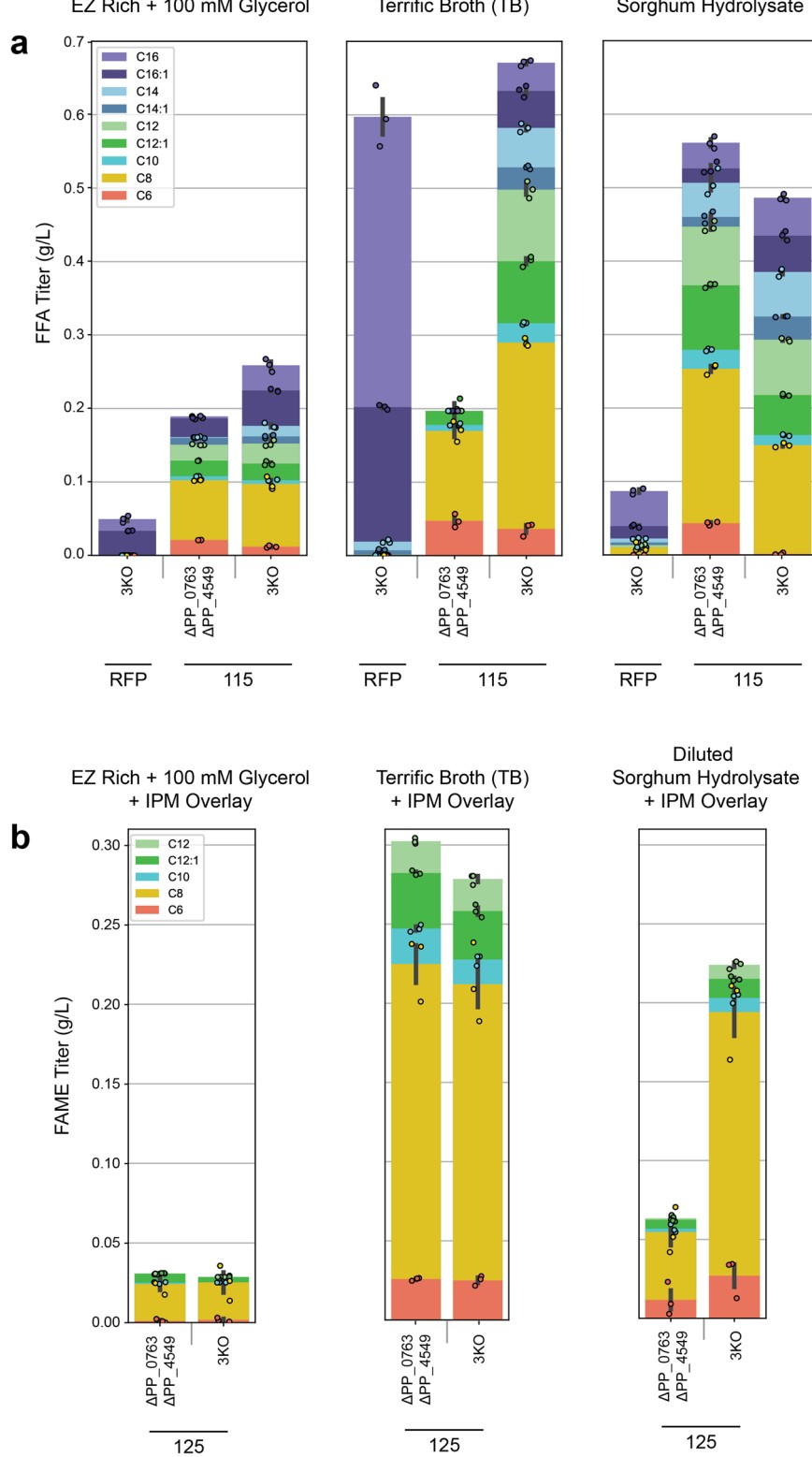

**Fig. 6 FFA and FAME production profile in various *P. putida* strains after growth in 250 mL shake flask cultures. a** FFA production in EZ Rich medium supplemented with 100 mM glycerol, TB, and diluted sorghum hydrolysate. **b** FAME production in EZ Rich medium supplemented with 100 mM glycerol, TB, and diluted sorghum hydrolysate. Values shown represent the mean of biologically independent samples ($n = 3$), and error bars show standard error of the mean.

Additionally, FFA and FAME titers from strains grown in diluted sorghum hydrolysate approached levels achieved in TB. In diluted sorghum hydrolysate, $\Delta PP\_0763$ $\Delta PP\_4549+115$ produced 561.1 mg/L total FFAs, the majority of which were medium-chain FFAs, while 3KO + 125 produced 224.0 mg/L total FAMEs. This further demonstrates the potential for using *P. putida* to sustainably produce oleochemicals from plant-derived carbon sources.

## Discussion

There has been a notable shift to plant and animal-derived oleochemicals as a substitute for petrochemicals. Microbial oleochemical production offers a versatile additional source of oleochemicals with unique properties such as branched-chains or shorter chain lengths than the predominant long-chain oleochemicals derived from plants and animals[7,8,10]. Production of these oleochemicals from cheap feedstocks, like plant biomass, is essential for these bioderived products to compete with their petroleum-derived counterparts. *P. putida* offers considerable advantages over *E. coli* and other more readily engineered microorganisms when it comes to producing oleochemicals from plant-derived feedstocks because of its ability to utilize and valorize lignocellulosic biomass[20]. The *P. putida* strains engineered in this work provide an important platform to produce FFA derived-oleochemicals from plant-derived feedstocks in this host.

Microbial oleochemical production benefits from the use of a host unable to catabolize fatty acids. Many oleochemical final products or intermediates can be degraded via β-oxidation[37], resulting in a decreased ability to accumulate the final product[27,38]. By removing several CoA ligases that initiate β-oxidation, we engineered *P. putida* strains unable to catabolize medium- and long-chain FFAs. Furthermore, the FFA product profile of several thioesterases was validated in *P. putida*, providing a reference for researchers that aim to produce a variety of FFA profiles in this organism.

The presence of CoA ligase paralogs with overlapping, but different, substrate ranges in *P. putida* is not surprising given its environmental niche as a saprophyte, where it likely encounters a diverse range of fatty acids. In this study, we took advantage of differences in CoA ligase substrate specificity to engineer two strains, $\Delta PP\_0763$ $\Delta PP\_4549$ and $\Delta PP\_4549-50$, that preferentially produce medium- or long-chain FFAs, respectively. This strategy may be leveraged in a variety of microbial hosts to control the final product range of oleochemicals by allowing the catabolism of unwanted FFA-like intermediates and final products. As an example, one could potentially limit the production of unwanted straight-chain FFAs in *E. coli* that have been engineered to produce branched-chain FFAs[39] by introducing an engineered CoA ligase that is able to ligate straight-chained FFAs while avoiding the desired branch-chained final products. The overexpression of strategic CoA ligases can be used in conjunction with other chain-length control measures such as engineered thioesterases[40], acyl-ACP:CoA transacylases[38], and engineered fatty acid synthases[41] to further control final product profiles.

Finally, this work provides *P. putida* strains that produce FFA titers of up to 670.9 mg/L total FFAs and 253.6 mg/L C8 FFA. These *P. putida* strains were also grown on sorghum hydrolysate, an industrially relevant plant-derived feedstock with lignocellulosic components that *P. putida* is capable of catabolizing, resulting in the production of medium-chain FFAs at titers greater than 450 mg/L. Additionally, the use of a medium-chain fatty acid methyltransferase demonstrated the utility of strains that could produce FFAs, resulting in a reported FAME titer of 302.4 mg/L. These titers fall below those achieved in *Escherichia coli*[25,32,38,40,42,43] and *Yarrowia lipolytica*[41,44,45], which indicates there is room for improvement by adopting the engineering strategies used in these classic oleochemical hosts as well as those used in alternative hosts such as *Saccharomyces cerevisiae*[46] and *Synechocystis* sp. PCC 6803[47] (Supplementary Tables 8 and 9). One such engineering strategy that has proven to be extremely powerful and led to an impressive FFA titer of 4.7 g/L by Wu et al.[42] is redox cofactor balancing, which aims to maintain an adequate supply of NADH reducing equivalents that are integral for fatty acid biosynthesis. Similarly, when producing methyl esters, a SAM dependent process, increasing the availability of SAM has led to increased titers[43,48]. Nevertheless, this work provides a solid framework for *P. putida* metabolic engineers that seek to use this advantageous host to produce oleochemicals from a variety of feedstocks.

## Methods

**Chemicals**. All chemicals were purchased from Sigma-Aldrich (United States) unless otherwise described.

**Plasmids and strains**. Plasmids and strains used in this study are listed in Supplementary Tables 10, 11. The plasmids and strains have been deposited in the public version of JBEI registry (http://public-registry.jbei.org) and are physically available from the corresponding author upon request. All plasmids were constructed via Gibson or Golden Gate Assembly using standard protocols. All heterologous genes were codon optimized for *P. putida* KT2440 and ordered as gBlocks from IDT.

**Culture conditions and plasmid transformations**. *E. coli* cultures used for cloning were grown in LB medium at 37 °C at 200 RPM, while *P. putida* cultures used for strain construction were grown in LB medium at 30 °C at 200 RPM. Selection markers were used at the following final concentrations: kanamycin (50 μg/mL), gentamicin (30 μg/mL), chloramphenicol (20 μg/mL), and sucrose (20% w/v). Episomal plasmids were transformed into *E. coli* and *P. putida* via electroporation. Briefly, to prepare competent cells 1 mL of overnight culture was centrifuged in 1.7 mL Eppendorf tubes at 21,300×g for 1 min. The supernatant was discarded, and pellets were washed with nanopure water three times. After the final wash the pellet was resuspended in 100 μL of nanopure water and 1 μL of purified plasmid was added to the competent cells. Cells were electroporated in a 0.1 cm cuvette at 1.8 kV. After electroporation, cells were resuspended in 500 μL LB and incubated at 37 °C at 200 RPM (*E. coli*) for one hour or 30 °C at 200 RPM (*P. putida*) for two hours before plating. Unless otherwise noted, cultures were grown in 50 mL 25 mm × 150 mm round-bottom borosilicate culture tubes.

**Growth media**. For FFA degradation assays EZ Rich medium (Teknova) supplemented with 10 mM glucose was used. For FFA and FAMEs production assays either EZ Rich medium supplemented with 100 mM glycerol, Terrific Broth medium (EMD Millipore), or diluted sorghum hydrolysate was used. Sorghum hydrolysate was diluted 4x with M9 minimal salts medium to a final 1x M9 salts concentration. After preparation, all media were sterile filtered, not autoclaved, through a 0.22 μm bottle-top filter.

**Sorghum hydrolysate production**. Sorghum (30% w/w of total weight), choline hydroxide (4% w/w of dry sorghum), lactic acid (6.3% of dry sorghum), acetic acid (5.8% of dry sorghum), and water were added to a 10 L Parr reactor with a total working mass of 3 kg. Lactic and acetic acid was added to react with choline hydroxide and produce cholinium lactate and cholinium acetate in situ as the pretreatment catalysts. The material was pretreated for three hours at 140 °C at 80 RPM. Subsequently, the pretreated slurry was cooled to 25 °C and the pH was adjusted to 5.0 with 6 N hydrochloric acid. A 9:1 ratio of cellulase CTec3 and hemicellulase HTec3 NS 22244 at 10 mg protein/g of biomass was added to the Parr reactor and enzymatic saccharification was conducted at 50 °C for 72 h at 80 RPM. After 72 h, the hydrolysate was centrifuged, and the supernatant was filtered through a 0.45um bottle-top filter. Following filtration, the pH of the hydrolysate was adjusted to 7.0 with 5 N NaOH. Finally, the hydrolysate was filtered once more through a 0.22μm bottle-top filter. The final hydrolysate contained 47.1 g/L glucose, 25.6 g/L xylose, 16.9 g/L acetic acid, and 17.3 g/L lactic acid.

**Generation of CoA ligase knockout mutants**. Deletion mutants in *P. putida* were constructed by allelic exchange as described previously[49]. Briefly, 1 kb homology regions upstream and downstream of the target gene, including the start and stop codons, were cloned into pMQ30. Plasmids were then conjugated into *P. putida* using *E. coli* S17 strains. Transconjugants were selected on gentamicin and chloramphenicol LB plates and then grown overnight in LB with no antibiotics. Overnight cultures were diluted 100x, 100 μL of which was plated on LB agar with

no NaCl that was supplemented with 10% (wt/vol) sucrose. Putative deletions were replica plated on LB agar with no NaCl supplemented with 10% (wt/vol) sucrose and LB agar with gentamicin. Colonies that grew on sucrose, but not gentamicin, were screened via PCR with primers flanking the target gene to confirm gene deletion.

**FFA degradation assays**. Tested strains were inoculated in 5 mL of EZ Rich medium supplemented with 10 mM glucose and grown overnight at 30 °C and 200 RPM. The cultures and a blank medium-only control were then back diluted 10x in EZ Rich medium supplemented with 10 mM glucose and spiked with 250 μM of each even-chained FFA between C6 and C16. Following this, cultures were grown at 30 °C for 48 h. The remaining FFAs were derivatized and quantified using GC/MS.

**Growth assays**. Carbon source growth assays were performed as previously described[27]. Briefly, overnight cultures were inoculated into 5 mL of LB medium from single colonies, and grown at 30 °C. These cultures were washed three times in carbon-free MOPS (morpholinepropanesulfonic acid) minimal medium, which is comprised of 32.5 μM CaCl$_2$, 0.29 mM K$_2$SO$_4$, 1.32 mM K$_2$HPO$_4$, 8 μM FeCl$_2$, 40 mM MOPS, 4 mM tricine, 0.01 mM FeSO$_4$, 9.52 mM NH$_4$Cl, 0.52 mM MgCl$_2$, 50 mM NaCl, 0.03 μM (NH$_4$)$_6$Mo$_7$O$_{24}$, 4 μM H$_3$BO$_3$, 0.3 μM CoCl$_2$, 0.1 μM CuSO$_4$, 0.8 μM MnCl$_2$, and 0.1 μM ZnSO$_4$. After washing, cultures were diluted 100x into 500 μL of MOPS minimal medium with 10 mM glucose or 10 mM FFA (C6-C12) was spiked in as a sole carbon source in 48-well plates (Falcon, 353072). Tergitol NP-40 was added to the medium to a final concentration of 1% (v/v) to help solubilize the FFAs. Plates were sealed with a gas-permeable microplate adhesive film (Breathe-Easy®, Sigma-Aldrich, Z380059), and then optical density at 600 nm (OD600) was monitored with a Biotek Synergy H1M at 30 °C for 48 h with fast continuous shaking (Agilent, Santa Clara, CA). To conduct growth assays on strains containing an IPM overlay, overnight cultures were diluted 100x into 5 mL of LB medium containing a 1:10 isopropyl myristate overlay in 50 mL 25 mm × 150 mm round-bottom borosilicate culture tubes and grown at 30 °C, 200 RPM, for 48 h. Periodically, 100 μL of the culture's aqueous phase was transferred to a clear bottom 96-well plate (VWR, CORN3912) and the OD600 was measured with a SpectraMax M2 plate reader.

**FFA and FAME production assays**. Strains were inoculated in 5 mL of medium containing kanamycin and grown overnight at 30 °C and 200 RPM. The media used for overnight cultures matched the media used for the production assay. Strains grown in sorghum hydrolysate had to be slowly adapted to this medium. First, strains were inoculated in a 20x dilution of hydrolysate in M9 minimal salts medium overnight, and subsequently back diluted into a 4x dilution of hydrolysate in M9 minimal salts and once again grown overnight. The next day, all overnight cultures were back diluted 10x into kanamycin-containing medium and grown at 30 °C for 4 h before adding arabinose to a final concentration of 0.2% v/v. A 1:10 isopropyl myristate overlay was added to FAME-producing cultures at the time of induction. Following induction with arabinose, cultures were grown for an additional 48 h before quantifying FFA and FAME production via GC/MS.

**FFA derivatization and quantification via GC/MS**. To derivatize FFAs in liquid culture, 300 μL of culture was added to a 1.5 mL screw cap microcentrifuge tube (VWR 16466-064) containing 15 μL of 40% tetrabutylammonium hydroxide. Each sample was spiked with a nonanoic acid internal standard to a final concentration of 250 μM. Then, 600 μL of dichloromethane containing 0.5% 2,3,4,5,6-penta-fluorobenzyl bromide (v/v) was added to each sample. The samples were incubated in an Eppendorf thermomixer at 50 °C and 1400RPM for 20 min. Following incubation, the samples were centrifuged at 21,300×g for 10 min and the organic layer was collected for GC/MS analysis. In parallel, blank medium samples containing 0 μM, 50 μM, 100 μM, 250 μM, 500 μM, or 750 μM mixture of each even-chained FFA between C6 and C16 were similarly spiked with a nonanoic acid internal standard and derivatized in triplicate.

Samples were run on an Agilent 6890 GC and Agilent 5973 MS using a 30 m × 0.250 mm × 0.25 μm HP-5MS column with a helium flow of 1.3 mL/min. The GC was run in splitless mode with a 1 μL sample injection and a constant inlet temperature and transfer line temperature of 250 °C. The oven starting temperature was 80 °C and held for 1 min. Then, the oven temperature was increased at a rate of 10 °C/min until 150 °C. Finally, the temperature was increased at a rate of 20 °C/min until 300 °C and held for 5 min. A solvent delay of 6.80 min was used and the MS was run in scan mode (50.0–350.0 amu). The extracted ion chromatograms at 181 m/z, a prominent fragment ion for pentafluorobenzyl esters, were analyzed via Chemstation Enhanced Data Analysis program. We identified compounds of interest by comparison with derivatized authentic standards.

Linear regression was used to generate a standard curve for each derivatized FFA using standards between 0 and 750 μM. More specifically, the extracted ion chromatogram peak area of the FFA of interest was normalized by the extracted ion chromatogram peak area of the derivatized nonanoic acid internal standard within each sample to generate an area ratio for each FFA present in the standard. Similarly, the average area ratios from biological replicates (n = 3) were calculated

for each FFA present in the samples and compared to a standard curve generated in the same media to determine the concentration of the FFA. If any calculated FFA area ratio within a sample fell outside the linear range of the standard curve, then a fresh aliquot of the sample was diluted 5x in media, spiked with internal standard, and re-analyzed to more accurately calculate the concentration of the FFA.

**FAME quantification via GC/MS**. All medium-chain FAMEs were found to evaporate from liquid media when maintained at 30 °C and 200RPM for 48 h. Therefore, a 1:10 isopropyl myristate (IPM) overlay was used in all samples from which we quantified FAME titers. Once ready for analysis, 1 mL from the surface of each sample was centrifuged at 21,300×g for 10 min. After centrifugation, 20 μL of the IPM layer was collected and diluted 10x with ethyl acetate spiked with 500 μM methyl nonanoate as an internal standard. In parallel, FAME standards were prepared by creating a 0 μM, 50 μM, 100 μM, 250 μM, 500 μM, 750 μM, 1 mM, 1.5 mM, and 2 mM mixture of each even-chained FAME between C6 and C16 in a 1:10 solution of IPM in ethyl acetate spiked with 500 μM methyl nonanoate as an internal standard in triplicate.

Samples were run on an Agilent 6890 GC and Agilent 5973 MS using a 30 m × 0.250 mm × 0.25 μm HP-5MS column with a helium flow of 1.3 mL/min. The GC was run in splitless mode with a 1 μL sample injection and a constant inlet temperature and transfer line temperature of 280 °C. The oven starting temperature was 50 °C and held for 1 min. Then, the oven temperature was increased at a rate of 10 °C/min until 160 °C. Finally, the temperature was increased at a rate of 30 °C/min until 250 °C and held for 5 min. A solvent delay of 4.8 min was used and the MS was run in scan mode (50.0–350.0 amu). The extracted ion chromatograms at 74 m/z, a prominent fragment ion for FAMEs, were analyzed via Chemstation Enhanced Data Analysis program. We identified compounds of interest by comparison with authentic standards.

Linear regression was used to generate a standard curve for each FAME using standards between 0 and 2 mM. More specifically, the extracted ion chromatogram peak area of the FAME of interest was normalized by the extracted ion chromatogram peak area of the methyl nonanoate internal standard within each sample to generate an area ratio for each FAME present in the standard. Similarly, the average area ratios from biological replicates (n = 3) were calculated for each FAME present in samples and compared to the standard curve to determine the concentration of the FAME.

Despite the use of an IPM overlay, we found that medium-chain FAMEs were still partially evaporating. To account for this, every production run had blank media controls (n = 3) that were spiked with a 500 μM FAME mixture before the addition of the IPM overlay. These controls were incubated alongside all biological samples and were similarly processed for FAME quantification. The average area ratios of each FAME in the spiked FAME controls were compared to the area ratios corresponding to 500 μM FAME in the standard curves and used to calculate an evaporation coefficient that accounts for the evaporation of FAMEs following the formula below (Eq. 1):

$$E = \frac{A_{standard}}{A_{spiked}} \qquad (1)$$

where $A_{standard}$ is the expected area ratio of a particular 500 μM FAME as calculated from the standard curve and $A_{spiked}$ is the observed area ratio of a particular 500 μM FAME that was spiked into a media control and incubated alongside biological samples. After calculating this coefficient for each FAME, this value was multiplied by the original calculated FAME concentration to determine a FAME concentration adjusted for evaporation.

**Theoretical yield calculations**. Using the COBRApy metabolic modeling package in Python[50], the maximum theoretical yield was calculated with the P. putida KT2440 model IJN1463[51] assuming no carbon flux to biomass. Individual thioesterase reactions were added for each free fatty acid analyte and methyl transferase reactions for the methyl ester analytes.

**Statistics and reproducibility**. All experiments involving the quantification of FFAs or FAMEs from bacterial liquid cultures were conducted with three biologically independent samples (n = 3) where each replicate was started from a different colony. The reported values were the mean of these biologically independent samples and error bars show the standard error of the mean.

**Reporting summary**. Further information on research design is available in the Nature Portfolio Reporting Summary linked to this article.

## Data availability
Plasmids and strains used in this study are listed in Supplementary Tables 10, 11. The plasmids and strains have been deposited in the public version of JBEI registry (http://public-registry.jbei.org) from which physical strains can be ordered. Source data for Figs. 2–6 and Supplementary Fig. 3 is found in Supplementary Data 1.

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

## Acknowledgements

We thank Chris Eiben for his advice on analytical chemistry, Ellen Zippi for her guidance on the manuscript, and Yuzhong Liu & Amin Zargar for valuable discussions. This work

was part of the DOE Joint BioEnergy Institute (https://www.jbei.org) supported by the U.S. Department of Energy, Office of Science, Office of Biological, and Environmental Research, and supported by the U.S. Department of Energy, Energy Efficiency and Renewable Energy, Bioenergy Technologies Office, through contract DE-AC02-05CH11231 between Lawrence Berkeley National Laboratory and the U.S. Department of Energy. L.E.V was supported by a National Science Foundation Graduate Research Fellowship, M.G.T. is a Simons Foundation Awardee of the Life Sciences Research Foundation, and M.M. was supported by the Amgen Scholars program. The laboratory of L.M.B. is partially funded by the Deutsche Forschungsgemeinschaft (DFG, German Research Foundation) under Germany's Excellence Strategy within the Cluster of Excellence FSC 2186 "The Fuel Science Center."

## Author contributions

Conceptualization: L.E.V., M.R.I., and J.D.K.; methodology: L.E.V., M.R.I., M.S., A.N.P., N.S., A.O., J.G., and J.D.K.; investigation: L.E.V., M.R.I., M.S., A.N.P., M.G.T., J.R., M.M., K.Y., N.S., and A.O.; writing – original draft: L.E.V; writing – review and editing: all authors; resources and supervision: P.M.S., L.M.B., J.G., and J.D.K.

## Competing interests

J.D.K. has financial interests in Amyris, Ansa Biotechnologies, Apertor Pharma, Berkeley Yeast, Demetrix, Lygos, Napigen, ResVita Bio, Zero Acre Farms, and Cyklos Materials. The other authors declare no competing interests.
