## [Peer Review File · Communications Biology]

Reviewers' comments:

Reviewer #4 (Remarks to the Author):

In this manuscript, the authors constructed of various CoA ligase knocking out mutant strains as well as expression of acyl-ACP thioesterase for accumulation of FFA. Furthermore, FAME production was also demonstrated. Overall, the manuscript is well-constructed and written. However, I found some issues to be addressed to improve the manuscript.

Several issues,

- I cannot find the growth curve of the WT and engineered strains. There is no yield data.
- The choice of IPM as the organic overlay is questionable. IPM exhibits the toxicity against gram-negative. The authors need to show the growth of engineered strains with and without the IPM.
- Further discussion on FAME difference between EZ Rich media and TB media need to be given. For example, why the WT strain produced higher FAME in EZ Rich media than in TB media, but the 3KO or other double mutant appear reversed? This phenotype cannot be explained based on the discussion given in page 8 line 216-223.

Overview

We appreciate the comments from the editor and reviewer #4 and have conducted an IPM toxicity study as well as addressed comments on the contents of the manuscript. Please find the point-by-point responses to the comments below.

Editor (Remarks to the Author):

From an editorial standpoint, we believe that the low titer should be more explicitly mentioned as a limitation of the study in the Discussion, and be further placed into the context of the Wu et al 2017 studies mentioned by the previous Reviewer #3.

We have revised the discussion section to more explicitly mention the low titer limitation and provide context regarding the different engineering strategies that could be adapted to improve FFA and FAMEs production, and highlighted the powerful method of redox cofactor balancing used by Wu et al. in 2017.

Reviewer #4 (Remarks to the Author):

In this manuscript, the authors constructed of various CoA ligase knocking out mutant strains as well as expression of acyl-ACP thioesterase for accumulation of FFA. Furthermore, FAME production was also demonstrated. Overall, the manuscript is well-constructed and written. However, I found some issues to be addressed to improve the manuscript.

1. I cannot find the growth curve of the WT and engineered strains. There is no yield data.

The growth curve and yield data are located in the Supplementary Information document, however they had not been mentioned in the main text. We have now fixed this by referencing these data in the main text of the manuscript.

2. The choice of IPM as the organic overlay is questionable. IPM exhibits the toxicity against gram-negative. The authors need to show the growth of engineered strains with and without the IPM.

We agree that it's important to demonstrate how using an IPM overlay affects the growth of strains. Despite reports of gram-negative toxicity, we did not observe any toxicity towards our strains when IPM was added as an overlay to liquid culture. To confirm this, we grew both the WT and 3KO strain with and without an IPM overlay in liquid culture and took OD600 measurements at several timepoints. These measurements had to be taken from the aqueous phase of the cultures because taking a measurement through the IPM layer drastically affects OD measurements making them inaccurate. As a result, one limitation of this experiment is that we could not

continuously take OD measurements but instead took measurements at 0, 6, 12, 24, and 48 hours after inoculation from saturated cultures. This toxicity study is now mentioned in the main text, the methods section, and the data are presented in the supplementary information.

This experiment demonstrated that the WT strain achieved a similar final OD600 with and without the use of an IPM overlay. Interestingly, when the 3KO strain is grown without an IPM overlay the OD600 decreases between the 24 hr timepoint and the 48 hr timepoint. This was also observed when the strain was grown on glucose as a sole carbon source (Supplementary Fig. 2). However, the use of an IPM overlay seems to prevent this decrease in OD600. One possible explanation for this is that the accumulation of endogenously-produced FFAs could be toxic to the cells after 48 hrs and an IPM overlay can relieve this stress by sequestering some of these FFAs.

3. Further discussion on FAME difference between EZ Rich media and TB media need to be given. For example, why the WT strain produced higher FAME in EZ Rich media than in TB media, but the 3KO or other double mutant appear reversed? This phenotype cannot be explained based on the discussion given in page 8 line 216-223.

We have revised our discussion to more clearly address and provide a hypothesis for this bifurcation in the pattern of FAMES titers among different strains. We hypothesize that the availability of the SAM cofactor plays a key role in the drastic difference in titer between strains grown in EZ Rich and TB media. SAM is produced from methionine and ATP precursors which are likely more readily available/produced from a very rich medium like TB. A significant clarifying observation is that FAMES titers are low across the board in strains grown in EZ Rich, and even though production is higher in the wild-type background than in the 3KO background, this difference is small in magnitude. If strains grown in EZ Rich indeed are struggling to generate sufficient amounts of SAM then it is possible that the wild-type strain, which grows faster and still has the ability to recycle FFAs for carbon and energy, can generate more SAM compared to 3KO and thus result in a slightly higher FAMES titer (~28 mg/L in WT vs. 8 mg/L in 3KO). Ultimately, we still consistently observe that to achieve high FAMES titers (> 100 mg/L) the strains must 1) be able to transiently accumulate FFAs (by having at least one CoA ligase KO) and 2) be grown on very rich media such as TB or sorghum hydrolysate.

REVIEWERS' COMMENTS:

Reviewer #4 (Remarks to the Author):

In accordance with reviewers' comments, the authors have successfully addressed the comments and revised manuscript well. I'd like to recommend that the revised manuscript can be accepted as is for publication.

Overview

We would like to thank the reviewers for their insightful feedback and comments which have helped improve the manuscript. Accordingly, we have made changes to the manuscript and detail the responses to individual comments below.

Reviewer #1 (Remarks to the Author):

Compliments to the authors for the original and interesting research work especially with regards to its impact on advancing a global bio-based economy.

The authors have clearly shown and implemented a sound metabolic engineering strategy in *P. putida* to increase the intra-cellular pool of short and medium chain fatty acids and derivatives, by selectively limiting the degradation of short-medium chain fatty acids through the double knock-out of two out of three endogenous fatty acyl CoA ligases with the simultaneous expression of *E. coli* thioesterase A. The authors additionally expressed a heterologous fatty acid methyltransferase for converting the free fatty acids to the higher value FAMES. The authors make use of a renewable waste feed-stock (lignocellulosic biomass) for the production of high value compounds, in this case short to medium chain fatty acids and their respective FAMES. Indeed short to medium chain length fatty acids and their derived oleo-chemicals are a class of rare compounds that are difficult to obtain from plant, plant-like or animal sources. Furthermore, such medium to short chain aliphatic compounds possess unique properties that outweigh the more abundant longer chain aliphatic compounds in applications ranging from the bio-fuel to the food (requires less energy for metabolism and absorption by humans) markets and industries.

Therefore, my recommendation would be to accept this article for publication in this journal.

Prof. Dr. Thomas Brück

Thank you for the kind comments; we are pleased to hear that the main objectives we aimed to accomplish in this work were conveyed clearly.

Reviewer #2 (Remarks to the Author):

In “Engineering *Pseudomonas putida* KT2440 for chain length tailored free fatty acid and oleochemical production”, Valencia and colleagues engineer *Pseudomonas putida* to produce medium chain length oleochemicals, including medium chain length free fatty acids and methyl esters. Overall, it is solid metabolic engineering work and a well written manuscript. My biggest concern is that *P. putida* is a well-established host that has been engineered for many applications, and oleochemical production has been targeted in many organisms,

and so this work may be more appropriate for a metabolic engineering-oriented journal.

I also wonder about the choice of *P. putida* for conversion of sorghum hydrolysate. While *P. putida* has been engineered to consume hemicellulosic sugars in the past, the wild type organism is not capable of utilizing them, meaning that the organism was only able to consume the glucose from the biomass, as well as likely consuming the acetate and lactate from the ionic liquid pretreatment.

One of the main advantages of using *P. putida* KT2440 for bioproduction from plant hydrolysates is that *P. putida* is a lignin-degrading microorganism. Lignin makes up roughly 25% of the dry biomass of sorghum so using a host that can utilize lignin as a source of carbon and energy is advantageous over other hosts that cannot degrade lignin but have been engineered to produce oleochemicals such as *E. coli* and *Y. lipolytica*. Although it was not quantified, the sorghum hydrolysate contains lignin monomers that can be utilized by *P. putida*. The main text has been slightly modified to clarify this point.

It is true that the wild-type KT2440 strain is not capable of utilizing the substantial amount of xylose present in the sorghum hydrolysate. Engineering these FFA accumulating strains to utilize xylose would be a valuable future step that would improve the ability to produce oleochemicals from hydrolysate. As these strains are the first strains unable to initiate β -oxidation of medium- and long-chain FFAs to our knowledge, we felt conducting these knockouts in a wild-type KT2440 background would provide the most utility to future researchers hoping to further engineer these strains.

Other comments:

1. The authors should report the yield of products. This is presumably not possible from the super rich TB medium, but it should at least be done for the defined EZ Rich medium. This is critical to understanding strain performance.

We have now calculated the yield of FFAs and FAMES for strains grown in EZ Rich medium supplemented with 100 mM glycerol. This was done using the COBRAPy metabolic modeling package in Python. The results are reported in Supplementary Table 2 and the methods section has been appended.

2. Along these lines, *P. putida* does not require medium supplementation with amino acids or nucleotides, which are components of EZ Rich medium. Why did the authors not test production in a true minimal (mineral salts) medium?

While *P. putida* could be grown on a true minimal salts medium, EZ Rich allows for the cells to have a similar growth rate to that found in rich media such as TB or plant hydrolysates. Since our back dilution, induction with arabinose, and GCMS analysis were all based on consistent timepoints, we felt EZ Rich would be the most appropriate

defined medium that we could use to easily compare to other rich undefined media as it limited variability in cell density and is more representative of industrial growth media.

3. Page 8, end of first paragraph: please state the value (or range of values) for the evaporation factor. There is a huge difference between 4% evaporation and 80% evaporation when considering the reliability of the analytics for this method. Also, I wonder if using an uninoculated control is actually a good proxy for what is happening during the bioconversion. I would imagine that gas stripping (from CO₂ production) could enhance FAME loss, while bacterial membranes could serve as a sink for FAMEs. The magnitude of these, and potentially other, factors could have a large impact on the reliability of the correction factor.

We agree, stating the evaporation factor for each FAME will help readers understand the extent of the evaporation. We have added Supplementary Table 3 which specifies the calculated evaporation factor for each FAME in each different condition. The evaporation factor ranged between 0.961 – 1.071 for C10 and C12 FAMEs across all media, which indicates that there was nearly no evaporation of these FAMEs from the IPM overlay and the titers remained relatively unchanged after the adjustment. For the C8 FAME there was minimal evaporation from the IPM overlay with an evaporation factor range of 1.058 – 1.135 in EZ Rich and TB (corresponding to ~5 – 12% evaporation from the IPM overlay) and an evaporation factor of 1.402 in sorghum hydrolysate (corresponding to ~29% evaporation from the IPM overlay). The sorghum hydrolysate evaporation factors were elevated across all FAMEs which is potentially due to a difference in the partition coefficient of these FAMEs in this media due to the high ionic strength of the media and the fact that this media was only used in the larger 250mL production runs. Finally, the evaporation factor was most significant for the C6 FAMEs with a range of 1.763 – 2.060 in EZ Rich and TB (corresponding to ~45 – 50% evaporation from the IPM overlay) and an evaporation factor of 4.478 in sorghum hydrolysate which corresponds to an approximately 78% loss from the IPM layer. Despite the variability in the evaporation factor of the C6 FAME, the final calculated titer for this particular FAME was relatively similar between the 250mL TB and sorghum hydrolysate production runs with values of 25.0 and 27.7 mg/L, respectively.

Inoculated controls were initially attempted, however, all strains at our disposal still contain the esterase *bioH* which has been shown in other organisms to hydrolyze methyl esters. As a result, when inoculated controls are used the evaporation factors are much larger and the FAME titers appear to be artificially inflated by this phenomenon. We believe the uninoculated control is the best strategy to accurately account for FAME evaporation from the overlay.

4. I am concerned about the reported FAA titer values in the presence of plasmid 125 because quantification relies on conversion of FAAs to FAMEs, which are the other product being made. Is there any way to actually know if FAAs are being made in these FAME-producing strains, or if it is an artifact of the analytics?

In our study we measured FFAs by converting the FFAs to pentafluorobenzyl esters rather than methyl esters, which helps us detect shorter chain FFAs. Despite this, we share the concern that it's possible methyl esters could be converted to pentafluorobenzyl esters during derivatization and lead to an elevated FFA quantification. However, there is evidence that the effect of this artifact is very limited. First, and perhaps most significantly, we only measured FFAs in samples without an overlay. In our experience, not having an overlay results in a nearly complete evaporation of C6-C12 FAMES from liquid media and therefore it is unlikely that there are many FAMES left solubilized in the media to confound the FFA quantification. Second, we observed that having a higher FAMES titer did not necessarily translate to having a higher FFA titer, as is seen when you compare the FAME titers to the FFA titers across the same strains in figures 5a and 5b. Finally, we observed that the FAME chain length profile of any particular strain is not necessarily reflected in the FFA chain length profile of the same strain; examples of this lack of correlation are seen in the product profiles in figures 5c and 5d (e.g., Δ PP_4549-50 produces virtually no C6, C8, or C10 FFAs but is shown to produce a significant amount of the respective C6, C8, and C10 FAMES). This supports the assumption that any FAMES being produced by a strain are not skewing the measurement of FFAs a noticeable amount. To clarify this, we modified the manuscript to point out that most FAMES evaporate from cultures without an overlay.

5. Page 10, top. The compounds in the 4X diluted hydrolysate should be listed here. Also, how clean is the hydrolysate? Does it exclusively contain glucose, xylose, lactate, and acetate as listed in the Methods? If not, what other components are in there?

The compounds found in the hydrolysate have now been listed in the main text of the manuscript. Aside from glucose, xylose, lactate, and acetate the hydrolysate also contains choline and lignin monomers. The amount of choline and lignin monomers in the final hydrolysate were not quantified but *P. putida* is capable of catabolizing choline and many lignin monomers.

6. I would have liked to see a direct comparison of titers (and yields) of the strains engineered here to the state of the art in other organisms, especially (but not limited to) E.coli and oleaginous yeast. Also, reference 40 is written in Chinese, and so it will be inaccessible to many readers.

We have now added a table comparing the titers of the strains in this study to the state of the art in other organisms in the supplementary information. Reporting the yields was unfortunately not possible as many studies did not do the calculations or were not comparable. Reference 40 has been edited.

7. Traditionally, a complete description of plasmid construction is required, rather than just pointing to an online database, but I guess that is up to the editor. Regardless, I would appreciate a more thorough description of the plasmids in

Table S3. For instance, what is the E.coli origin of replication, P. putida origin of replication (when appropriate), selectable marker(s), oriT? sacB? Etc.

The plasmids were constructed using a variety of methods including golden gate assembly and Gibson assembly. Additionally, the DNA used to construct the plasmids was either generated via PCR or synthesized from IDT. We have added more columns to Table S4 to provide more details regarding the different elements of the plasmids. The JBEI public registry provides a convenient database where readers can get detailed descriptions of the plasmid components, download annotated plasmid maps, and order physical samples of strains and plasmids of interest. However, we agree that a more detailed table in the supplementary information file helps users more quickly understand the structure of the expression plasmids used.

8. Page 13 in “Carbon source growth assays”: describe the MOPS minimal medium. Also, what is the reason for use of Tergitol? And because aeration is important for P. putida, which “gas-permeable microplate adhesive film” was used? I did not find a product with that name on the VWR website.

The methods section has now been modified to give a description of the MOPS minimal medium and specify that the part number of the adhesive film (Breathe-Easy®, Sigma-Aldrich, Z380059). Tergitol NP-40 is used to help solubilize the FFAs that are being used as a carbon source and we verified that *P. putida* is not capable of growing on Tergitol NP-40 as a sole carbon source in MOPS minimal medium. This clarification has also been added to the methods section.

9. Page 14, top and bottom: I presume it is a 30 cm column, not a 30 m column? Also, spell out “temperature” twice (each) in the first and last full paragraphs.

Thank you, we corrected those instances where we spelled temp instead of temperature. The Agilent HP-5MS column is actually 30 meters which helps provide an adequate amount of separation for gas chromatography purposes.

10. JDK is the only one declaring a potential conflict of interest. Has any intellectual property been filed on this work that should also be noted?

No IP has been filed on this work.

Reviewer #3 (Remarks to the Author):

Valencia a colaborators engineered *P. putida* for the production of medium and long free fatty acids and their ester forms.

Despite the considerable amount of work presented in this study the titers are rather low and are behind those obtained in *E. coli* > 4,500 mg/L. See the

following references:

Wu et al 2017., Met. Eng. (3,000 mg/L)

wu et al 2017 (<https://doi.org/10.1016/j.ymben.2017.11.001>) 4,500 mg/L FFA

While our titers do fall behind the impressive work done by Wu et al., it is still notable that this work opens the doors to the production of many oleochemicals in *P. putida* which is able to utilize recalcitrant carbon sources that *E. coli* can't utilize such as lignin monomers. Our strains provide a starting point that future researchers could further engineer (e.g., incorporation of formate dehydrogenase system for regeneration of NADH, overexpression of acetyl-CoA carboxylase, etc.) to improve FFA and FAME production in this advantageous host. Additionally, to our knowledge, this work includes the first *in vivo* production of methyl hexanoate and methyl octanoate in an engineered host. This confirms the hypothesis that the methyltransferase MmFAMT can produce medium-chain FAMES if expressed in a host that produces medium-chain FFAs; this information may be useful to engineers aiming to biologically produce FAMES.

Overview

We appreciate the comments from the editor and reviewer #4 and have conducted an IPM toxicity study as well as addressed comments on the contents of the manuscript. Please find the point-by-point responses to the comments below.

Editor (Remarks to the Author):

From an editorial standpoint, we believe that the low titer should be more explicitly mentioned as a limitation of the study in the Discussion, and be further placed into the context of the Wu et al 2017 studies mentioned by the previous Reviewer #3.

We have revised the discussion section to more explicitly mention the low titer limitation and provide context regarding the different engineering strategies that could be adapted to improve FFA and FAMEs production, and highlighted the powerful method of redox cofactor balancing used by Wu et al. in 2017.

Reviewer #4 (Remarks to the Author):

In this manuscript, the authors constructed of various CoA ligase knocking out mutant strains as well as expression of acyl-ACP thioesterase for accumulation of FFA. Furthermore, FAME production was also demonstrated. Overall, the manuscript is well-constructed and written. However, I found some issues to be addressed to improve the manuscript.

1. I cannot find the growth curve of the WT and engineered strains. There is no yield data.

The growth curve and yield data are located in the Supplementary Information document, however they had not been mentioned in the main text. We have now fixed this by referencing these data in the main text of the manuscript.

2. The choice of IPM as the organic overlay is questionable. IPM exhibits the toxicity against gram-negative. The authors need to show the growth of engineered strains with and without the IPM.

We agree that it's important to demonstrate how using an IPM overlay affects the growth of strains. Despite reports of gram-negative toxicity, we did not observe any toxicity towards our strains when IPM was added as an overlay to liquid culture. To confirm this, we grew both the WT and 3KO strain with and without an IPM overlay in liquid culture and took OD600 measurements at several timepoints. These measurements had to be taken from the aqueous phase of the cultures because taking a measurement through the IPM layer drastically affects OD measurements making them inaccurate. As a result, one limitation of this experiment is that we could not

continuously take OD measurements but instead took measurements at 0, 6, 12, 24, and 48 hours after inoculation from saturated cultures. This toxicity study is now mentioned in the main text, the methods section, and the data are presented in the supplementary information.

This experiment demonstrated that the WT strain achieved a similar final OD600 with and without the use of an IPM overlay. Interestingly, when the 3KO strain is grown without an IPM overlay the OD600 decreases between the 24 hr timepoint and the 48 hr timepoint. This was also observed when the strain was grown on glucose as a sole carbon source (Supplementary Fig. 2). However, the use of an IPM overlay seems to prevent this decrease in OD600. One possible explanation for this is that the accumulation of endogenously-produced FFAs could be toxic to the cells after 48 hrs and an IPM overlay can relieve this stress by sequestering some of these FFAs.

3. Further discussion on FAME difference between EZ Rich media and TB media need to be given. For example, why the WT strain produced higher FAME in EZ Rich media than in TB media, but the 3KO or other double mutant appear reversed? This phenotype cannot be explained based on the discussion given in page 8 line 216-223.

We have revised our discussion to more clearly address and provide a hypothesis for this bifurcation in the pattern of FAMES titers among different strains. We hypothesize that the availability of the SAM cofactor plays a key role in the drastic difference in titer between strains grown in EZ Rich and TB media. SAM is produced from methionine and ATP precursors which are likely more readily available/produced from a very rich medium like TB. A significant clarifying observation is that FAMES titers are low across the board in strains grown in EZ Rich, and even though production is higher in the wild-type background than in the 3KO background, this difference is small in magnitude. If strains grown in EZ Rich indeed are struggling to generate sufficient amounts of SAM then it is possible that the wild-type strain, which grows faster and still has the ability to recycle FFAs for carbon and energy, can generate more SAM compared to 3KO and thus result in a slightly higher FAMES titer (~28 mg/L in WT vs. 8 mg/L in 3KO). Ultimately, we still consistently observe that in order to achieve high FAMES titers (> 100 mg/L) the strains must 1) have the ability to transiently accumulate FFAs (by having at least one CoA ligase KO) and 2) must be grown on very rich media such as TB or sorghum hydrolysate.